# Multiple Postoperative Neuropathies in a Morbidly Obese Patient Undergoing Open Total Hysterectomy: A Case Report

**DOI:** 10.3390/reports8040245

**Published:** 2025-11-25

**Authors:** Mugi Kawafune, Joho Tokumine, Harumasa Nakazawa, Kiyoshi Moriyama

**Affiliations:** Department of Anesthesiology, Kyorin University School of Medicine, 6-20-2 Sinkawa, Mitaka 181-8611, Tokyo, Japan; mugi@ks.kyorin-u.ac.jp (M.K.); jtokumine@ks.kyorin-u.ac.jp (J.T.); mokiyo@ks.kyorin-u.ac.jp (K.M.)

**Keywords:** morbid obesity, postoperative neuropathy, hysterectomy, femoral nerve injury, lateral femoral cutaneous nerve, perioperative positioning

## Abstract

Background and Clinical Significance: Morbid obesity is a recognized risk factor for perioperative neuropathies, but simultaneous involvement of multiple peripheral nerves is rare. Case presentation: We report a 49-year-old woman (BMI 45) who underwent open total hysterectomy under combined general and epidural anesthesia in the supine position with Trendelenburg tilt. Despite preoperative positioning simulation, she developed postoperative numbness of the left fingers and bilateral upper limb weakness, followed by bilateral lower limb weakness and severe right thigh pain. Imaging excluded epidural hematoma. Iliopsoas weakness persisted despite epidural discontinuation. Rehabilitation and pregabalin partially improved symptoms, but sensory deficits and gait disturbance remained. At long-term follow-up, she was able to ambulate with a Lofstrand crutch, although right thigh numbness persisted. Two years postoperatively, diabetes mellitus was diagnosed. Conclusions: Morbid obesity may predispose to multiple perioperative neuropathies through mechanical compression and metabolic vulnerability. Careful intraoperative reassessment of pillow height, limb position, and retractor placement, combined with early recognition and multidisciplinary management, is essential to prevent neuropathy and optimize recovery.

## 1. Introduction and Clinical Significance

A recent study showed that morbid obesity is a risk factor for neuropathy [1,2]. Furthermore, morbid obesity has been reported to increase perioperative complications, including neuropathies caused by mechanical compression or malpositioning during surgery [3,4]. Although several reports describe isolated postoperative femoral or brachial plexus neuropathy, the rare occurrence of multiple neuropathies affecting both the upper and lower extremities simultaneously has been documented [5,6].

Here, we present a case of a morbidly obese patient who developed multiple postoperative neuropathies after an open total hysterectomy, and we discuss potential mechanisms and preventive strategies.

## 2. Case Presentation

A 49-year-old woman (height 161 cm, weight 116 kg, body mass index 45) with no significant medical history was scheduled for open total hysterectomy and bilateral salpingo-oophorectomy for adenomyosis. She had never been diagnosed with diabetes, and her preoperative fasting blood glucose level was 107 mg/dL, indicating no hyperglycemia. HbA1c was not measured preoperatively. On the day before surgery, a positioning simulation was performed to confirm the ramp position for tracheal intubation. No abnormalities in mouth opening or neck flexion/extension were observed, and there were no symptoms of nerve compression in the position.

General anesthesia combined with epidural anesthesia was planned. The epidural catheter was inserted at the T11/12 interspace on the first attempt without difficulty while the patient was in a sitting position. General anesthesia was induced with total intravenous anesthesia using target-controlled infusion of propofol (3.0 μg/mL), continuous infusion of remifentanil (0.1 μg/kg/min), and rocuronium bromide (70 mg). Mask ventilation was easily performed in the ramp position, and tracheal intubation was smoothly completed using a McGRATH™ with Blade Size 3 (Medtronic Japan Co., Ltd., Tokyo, Japan) without excessive neck extension. Anesthesia was maintained with target-controlled infusion of propofol and continuous infusion of remifentanil, monitored with Bispectral Index (BIS). Neuromuscular blockade was monitored by train-of-four (TOF), and rocuronium was administered as needed to keep a TOF count of 0. The arterial line was successfully placed in the left radial artery on the first attempt by palpation method. The patient was positioned supine with the legs abducted without flexing the hips or knees, ensuring that the upper limbs were abducted less than 90°. During the procedure, the patient was repositioned from supine to Trendelenburg tilt to improve visualization during the latter part of the surgery. The surgeon unintentionally applied pressure to the hand rest, causing excessive abduction of the left arm. The nurse promptly adjusted the arm position whenever the surgeon pressed and abducted the patient’s arm. After discontinuing propofol and remifentanil at the end of surgery, neuromuscular blockade was reversed with sugammadex. The patient’s emergence from anesthesia and extubation proceeded smoothly. The surgery lasted 4 h and 30 min, with an anesthesia duration of 5 h and 35 min.

Immediately after emerging from anesthesia, numbness was observed in the left third to fifth fingers. An ultrasound exam was conducted to rule out nerve injury caused by hematoma from arterial catheterization. It found no hematoma around the nerve, and no signs of nerve injury such as swelling were observed in the median nerve or the superficial branch of the radial nerve. Neurological examination revealed bilateral upper-limb weakness. Manual muscle testing (MMT) showed deltoid and biceps brachii strength of 2/5 on both sides, with triceps strength preserved at 4/5. Distal weakness was more pronounced on the left (wrist extensors 3/5, wrist flexors 3/5), compared with the right (4/5 for both). Sensory disturbance was present below the acromion bilaterally. The attending physician consulted an orthopedic surgeon. The exam showed negative Tinel’s and Phalen’s tests, indicating no entrapment neuropathy at the carpal tunnel, but spondylosis was suspected. A magnetic resonance imaging (MRI) scan was scheduled for the following day. However, symptoms improved by postoperative day (POD) 1. On POD 1, bilateral lower limb weakness occurred. Reducing the continuous epidural infusion (0.125% levobupivacaine) from 4 to 2 mL/h alleviated muscle weakness. Yet, electric shock-like pains and numbness appeared in the right lateral thigh. Lumbar MRI showed no epidural hematoma but revealed mild L4/5 disk herniation and bilateral foraminal stenosis (Figure 1). Discontinuing the epidural infusion improved motor function, but MMT indicated iliopsoas muscle weakness. Detailed muscle testing showed right iliopsoas strength of 1/5, right hamstring strength of 5/5, right quadriceps strength of 4/5, right tibialis anterior strength of 4/5, and right gastrocnemius strength of 5/5. The right patellar tendon reflex was slightly diminished, while the Achilles tendon reflex was preserved. In addition, sensory examination revealed decreased sensation in the right lateral thigh, rated 0–1/10. Pregabalin was started for neuropathic pain. After beginning pregabalin, the shock-like pain disappeared; however, dull sensation, numbness, and muscle weakness in the right thigh persisted. The patient was discharged on POD 10 using axillary crutches. At POD 150, sensory disturbances and gait issues persisted, though she could perform daily activities with medication adjustments and outpatient rehabilitation. Two years later, she transitioned from axillary crutches to Lofstrand (forearm) crutches. Subsequently, she was diagnosed with diabetes mellitus with an HbA1c of 8.2%.

## 3. Discussion

This case highlights the rare occurrence of multiple postoperative neuropathies in a morbidly obese patient. Upper limb symptoms were transient and may have been exacerbated by cervical spondylosis and pillow height rather than brachial plexus traction. The lower limb weakness was most likely caused by the pharmacologic effect of the epidural local anesthetic rather than by a complication of the epidural procedure itself, as motor strength improved after discontinuation of the infusion and MRI showed no evidence of an epidural hematoma. The distribution of sensory deficits suggested femoral or lateral femoral cutaneous nerve involvement, plausibly due to retractor-induced compression during open hysterectomy [7].

### 3.1. Mechanical Factors

Retractor-induced femoral neuropathy is a recognized complication of open gynecologic surgery, especially when self-retaining retractors are used [8]. In this case, a self-retaining retractor was carefully applied after laparotomy and remained in place during the procedure. Despite careful placement, prolonged retraction may have compressed the psoas major muscle and femoral nerve, resulting in anterior thigh sensory disturbance and iliopsoas weakness. Morbid obesity likely further increased susceptibility by narrowing the pelvic space and requiring greater retraction force [8]. Periodic release and reassessment of retractor position are recommended for essential preventive strategies [8].

Morbid obesity increases susceptibility to nerve compression due to increased soft tissue pressure and challenges in achieving and maintaining optimal positioning [3]. In this case, to secure the surgical field, we had to adopt the Trendelenburg position during the latter part of the surgery. A steep Trendelenburg position is known to carry a risk of nerve injury [4]. Positioning devices may specifically cause brachial plexus injury. Downward pressure on the shoulder in a steep Trendelenburg position can cause stretch injury to the brachial plexus. However, in this case, the Trendelenburg position was only slightly lowered, and no shoulder immobilization device was used. Additionally, care was taken to prevent excessive abduction of the upper limbs. Unfortunately, we did not pay much attention to the height of the pillow. Distracted by the ramp position for intubation, we failed to perform a preoperative positioning simulation regarding the pillow height after returning the patient to the horizontal supine position. Obesity has been found to be associated not only with lumbar spine disorders but also with cervical spine disorders [9]. Therefore, when performing surgery on morbidly obese patients, even if they are asymptomatic, the procedure should be conducted with the possibility of coexisting spinal disorders in mind.

### 3.2. Metabolic/Inflammatory Factors

Obesity may contribute to neuropathy not only through mechanical compression of peripheral nerves but also via chemical mechanisms such as chronic low-grade inflammation. Obesity induces macrophage infiltration into adipose tissue and dysregulated adipokine secretion, leading to persistent elevation of inflammatory cytokines such as interleukin-6 (IL-6), tumor necrosis factor-alpha (TNF-α), and C-X-C motif chemokine ligand 10 (CXCL10) [2]. These mediators promote microvascular dysfunction and oxidative stress in peripheral nerves, ultimately resulting in axonal damage and distal sensorimotor polyneuropathy [2]. In a large population-based study (KORA F4/FF4 cohort), circulating levels of CCL7, CXCL10, and DNER partially mediated the association between obesity and incident distal sensorimotor polyneuropathy, supporting the hypothesis that chronic subclinical inflammation is a mechanistic link between obesity and neuropathy [2].

Previous reports have linked obesity, prolonged surgery, and self-retaining retractors with postoperative neuropathy. The ASA Practice Advisory also emphasizes the importance of proper intraoperative positioning, the use of protective padding, and careful placement of equipment to avoid excessive pressure or stretch on peripheral nerves [10]. Although these recommendations were fully considered and applied in this case, perioperative neuropathies may still occur in challenging clinical situations such as morbid obesity and prolonged retraction. This case underscores that even with preoperative positioning simulation, intraoperative reassessment and adjustment of pillow height, limb position, and retractor placement are essential. Furthermore, early recognition of neurological symptoms, prompt imaging, and multidisciplinary management (orthopedics, neurology, rehabilitation) are crucial to optimize recovery.

### 3.3. Differentiation from Postoperative Inflammatory Neuropathy

Postoperative inflammatory neuropathy has been described as an immune-mediated cause of perioperative neuropathy, characterized by focal, multifocal, or diffuse neurological deficits developing within 30 days after surgery [6]. These neuropathies are associated with axonal injury, T2 hyperintensity or enlargement of roots or plexuses on MRI, and epineurial perivascular lymphocytic inflammation or microvasculitis on nerve biopsy. They often present with severe neuropathic pain and weakness extending beyond a single mechanical compression site, and may respond to corticosteroid therapy.

In the present case, several clinical features were more consistent with mechanical, compressive neuropathy rather than postoperative inflammatory neuropathy. The onset of symptoms was closely related to intraoperative events: upper-limb symptoms appeared immediately after emergence from anesthesia, and lower-limb weakness developed on postoperative day 1, corresponding to the use of a self-retaining retractor during prolonged open pelvic surgery and the Trendelenburg position. The distribution of deficits was largely confined to the femoral and lateral femoral cutaneous nerve territories, supported by a diminished patellar tendon reflex with preserved Achilles tendon reflex and a well-localized sensory deficit in the lateral thigh, rather than a multifocal or migratory pattern. In addition, lumbar MRI did not show root or plexus enlargement or features suggestive of inflammatory neuropathy, and no progression of weakness occurred beyond the early postoperative period. Gradual improvement with discontinuation of epidural infusion, analgesic adjustment, and rehabilitation—without the use of immunotherapy—also favored a mechanical rather than inflammatory etiology.

Although electromyography (EMG) was not performed in this case, EMG can be valuable in localizing the site of nerve injury and distinguishing mechanical compression from inflammatory neuropathies. Incorporating EMG in future similar cases may therefore improve diagnostic accuracy.

Taken together, although postoperative inflammatory neuropathy cannot be completely excluded in the absence of nerve biopsy, the temporal course, anatomical distribution, imaging findings, and response to conservative management strongly suggest that mechanical factors—particularly retractor-related compression in the setting of morbid obesity and underlying spinal pathology—were the predominant contributors to the patient’s neuropathies.

### 3.4. Rehabilitation for Postoperative Neuropathies

Rehabilitation plays a critical role in the recovery process of patients with perioperative neuropathies. Although the evidence in surgical populations is limited, several studies on peripheral nerve injury suggest that targeted rehabilitation approaches may promote functional improvement and facilitate nerve regeneration. Exercise-based therapy and neuromuscular re-education have been shown to enhance axonal regeneration, prevent disuse atrophy, and improve sensorimotor integration, particularly when initiated early in the recovery period. Systematic reviews have demonstrated that physical training and adjunctive modalities, such as electrical stimulation, can support microvascular perfusion and reduce the negative effects of denervation on muscle tissue [11].

In addition to exercise therapies, multimodal rehabilitation—combining strengthening exercises, gait training, sensory re-education, and neuromodulatory interventions such as TENS or peripheral magnetic stimulation—has been reported to yield functional benefits in patients with neuropathic deficits. Case-based evidence suggests that such multimodal programs can improve pain, enhance motor recovery, and assist patients in regaining mobility after peripheral nerve injury.

Functional outcomes following neuropathy are also influenced by several modifiable factors. Early recognition of neurological deficits, timely referral to rehabilitation specialists, and consistent participation in therapy are associated with improved recovery trajectories [12]. Furthermore, comorbidities such as obesity and metabolic dysfunction may impede nerve repair by promoting chronic low-grade inflammation; thus, multidisciplinary management, including metabolic optimization and physical conditioning, may further support neurological recovery.

Although standardized rehabilitation protocols for perioperative compressive neuropathies have not yet been established, current evidence suggests that individualized, early-initiated, and multimodal rehabilitation strategies may contribute significantly to recovery. Incorporating structured rehabilitation into postoperative management should therefore be considered an essential component of the care of patients with perioperative neuropathies.

## 4. Conclusions

Preoperative positioning simulation is important but does not fully eliminate the risk of perioperative neuropathies in morbidly obese patients. Careful intraoperative adjustment of pillow height, limb positioning, and retractor placement, along with postoperative vigilance and multidisciplinary management, is essential for prevention and recovery.

## Figures and Tables

**Figure 1 reports-08-00245-f001:**
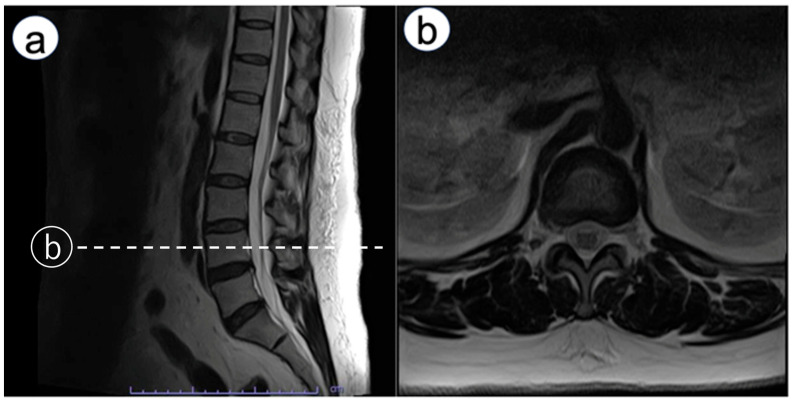
Lumbar magnetic resonance imaging. Sagittal and axial MRI images showing no evidence of epidural hematoma, with mild L4/5 disk herniation and bilateral foraminal stenosis. (**a**): sagittal view, (**b**): axial view (L4 level, dotted line).

## Data Availability

The original contributions presented in this study are included in the article. Further inquiries can be directed to the corresponding author.

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
