# Peer review of "Multiple Postoperative Neuropathies in a Morbidly Obese Patient Undergoing Open Total Hysterectomy: A Case Report"

_reports, 2025, doi:10.3390/reports8040245_

Round 1

Reviewer 1 Report

Comments and Suggestions for Authors

Multiple postoperative neuropathies in a morbidly obese patient undergoing open total hysterectomy: a case report

The case report is clear and concisely written. The authors have covered the topic well and explained  the potential rationale for the persistent injury. The authors have also provided useful suggestions for future situations as the one described. The authors have taken an unbiased view in the report.

The case report is sufficient for dissemination. 

Author Response

Comment: The case report is clear and concisely written. The authors have covered the topic well and explained  the potential rationale for the persistent injury. The authors have also provided useful suggestions for future situations as the one described. The authors have taken an unbiased view in the report.

The case report is sufficient for dissemination.

Response:  Thank you very much for your positive and encouraging comments. We appreciate your thoughtful review.

Reviewer 2 Report

Comments and Suggestions for Authors

The authors present a case of multiple postoperative neuropathies in a morbidly obese patient undergoing open total hysterectomy. The manuscript is well written and introduced the topic very well. The authors also described the intraoperative setting, environment, and events in great detail. However, a few points need to be addressed.  

  1. Could the authors describe the neurological exam in more details and provide detailed muscle strength testing?  
  2. Was an EMG study performed for the patient? And if yes, could it be added to the case presentation section? 
  3. The authors suggested the possibility of femoral nerve palsy or involvement. Did the patient have an absent knee jerk on the same side? This may also assist readers in reaching that conclusion.  
  4. It would be interesting if the authors can add a section in the discussion dedicated to the natural history of post-operative compressive neuropathies and if there are specific rehabilitation techniques that might help in the management.  
  5. Are there any societal guidelines from anesthesiology societies on prevention of such postoperative complications?  
  6. The author may also discuss the entity of “post operative inflammatory neuropathy” that may mimic multiple compression neuropathies (10.1093/brain/awq252) 

Author Response

Comment: The authors present a case of multiple postoperative neuropathies in a morbidly obese patient undergoing open total hysterectomy. The manuscript is well written and introduced the topic very well. The authors also described the intraoperative setting, environment, and events in great detail. However, a few points need to be addressed.  

Response: We appreciate the reviewer’s thorough evaluation and constructive feedback. Our responses to each point are detailed below.

Comments 1: Could the authors describe the neurological exam in more details and provide detailed muscle strength testing?  

Response 1: We recognize the importance of providing a clearer neurological assessment. Accordingly, we have expanded the description of the neurological examination and added detailed manual muscle testing (MMT) findings to illustrate both upper- and lower-limb deficits. These revisions appear in the updated “Case Presentation” section (lines 74-78, 88-92).

Comments 2: Was an EMG study performed for the patient? And if yes, could it be added to the case presentation section? 

Response 2: Thank you for your thoughtful comment. An EMG study was not conducted in this case. We agree that electrophysiological testing could have offered more precise localization of nerve involvement. To address this limitation, we have added a statement to the Discussion section noting the value of EMG in similar clinical situations (lines 186-189).

Comments 3: The authors suggested the possibility of femoral nerve palsy or involvement. Did the patient have an absent knee jerk on the same side? This may also assist readers in reaching that conclusion.  

Response 3: We agree with your insightful comment. In the revised manuscript, we have added the tendon reflex findings. The right patellar tendon reflex was diminished, while the Achilles tendon reflex was preserved, suggesting that the neurological deficit was relatively localized to the femoral nerve territory. These findings have now been incorporated into the Case Presentation section (lines 90-92).

Comments 4: It would be interesting if the authors can add a section in the discussion dedicated to the natural history of post-operative compressive neuropathies and if there are specific rehabilitation techniques that might help in the management.  

Response 4: We agree that such a section would strengthen the manuscript. A new subsection has been added to the Discussion outlining the natural history of postoperative compressive neuropathies and summarizing relevant rehabilitation approaches that may assist recovery (lines 197-225).

Comments 5: Are there any societal guidelines from anesthesiology societies on prevention of such postoperative complications?  

Response 5: Thank you for this important comment. Yes, guidelines from anesthesiology societies do exist. In particular, the American Society of Anesthesiologists (ASA) Practice Advisory provides recommendations for the prevention of perioperative peripheral neuropathies, including appropriate patient positioning, use of protective padding, and careful placement of equipment to avoid undue pressure or stretch on peripheral nerves. We have incorporated this information into the revised Discussion section.

In addition, we noted that although these preventive strategies were fully implemented in our patient, perioperative neuropathies can still occur in challenging clinical settings such as morbid obesity and prolonged retraction. This point has also been added to highlight the limitations of prevention even when guideline-based precautions are followed (lines 154-159).

Comments 6: The author may also discuss the entity of “post operative inflammatory neuropathy” that may mimic multiple compression neuropathies (10.1093/brain/awq252) 

Response 6: We have incorporated the reviewer’s suggestion by adding a discussion of postoperative inflammatory neuropathy, including its clinical characteristics and its potential to mimic multiple compression neuropathies. The referenced article has been cited, and the content has been added to the Discussion section (lines 164-195).

Reviewer 3 Report

Comments and Suggestions for Authors

The authors describe a fascinating case of multiple neuropathies following gynecologic surgery in a patent with obesity. The describe the details of the case well and offer several explanations as to the causes of the symptoms as well as the potential for recovery. This case report will be of interest, especially within the surgical community, as an illustrative case of potential complications. However, there were several questions that I feel, if addressed, could strength the report.

1. Could you please clarify if there had been any testing of blood sugar (a1c, fasting glucose, oral glucose tolerance testing, etc) prior to the surgery? Pre-diabetes can be associated with the typical length-dependent axonal neuropathy and may therefore place the patient at greater risk of nerve injury, as well.

2. You describe the surgeon placing their hand on the arm of the patient, leading to ‘excessive abduction of the arm’ (page 2 line 61-62). Could you please clarify which arm? I suspect it would be the left, give discussion of possible brachial plexus injury, but it is not explicitly stated.

3. Was the left radial arterial line placed under ultrasound guidance, or by palpation alone? You confirm that there was no clear vascular injury by ultrasound (page 2 line 68), but there is no mention of nerve (superficial radial/median) integrity. Was this checked?

4. You mention a negative Tinel’s and Phalen exonerates carpal tunnel syndrome (page 2, line 72), but the sensitivity of these two maneuvers is quite low at 34-50% and 67-73%, respectively (see doi:10.1001/jama.2022.8481 AND doi:10.1093/ptj/pzad029). Was a nerve conduction/EMG completed?

5. In Figure 1, please indicate (by dotted line and text) which level is shown in axial view.

6. In your discussion (page 3, line 99) you mention that symptoms were not due to the epidural, but then say that weakness improved after discontinuation. While I agree that there was evidence of a hematoma from the epidural anesthesia causing the symptoms, I would suggest clarifying this in the text. Clearly, if her weakness improved by stopping the anesthesia, then the anesthesia was causing the weakness (as expected), just not due to a hematoma.

Author Response

Comments: The authors describe a fascinating case of multiple neuropathies following gynecologic surgery in a patent with obesity. The describe the details of the case well and offer several explanations as to the causes of the symptoms as well as the potential for recovery. This case report will be of interest, especially within the surgical community, as an illustrative case of potential complications. However, there were several questions that I feel, if addressed, could strength the report.

Response: We appreciate the reviewer’s careful reading and constructive feedback. Below, we provide point-by-point responses.

Comments 1: Could you please clarify if there had been any testing of blood sugar (a1c, fasting glucose, oral glucose tolerance testing, etc) prior to the surgery? Pre-diabetes can be associated with the typical length-dependent axonal neuropathy and may therefore place the patient at greater risk of nerve injury, as well.

Response 1: Thank you for this important comment. The patient had never been suspected of having diabetes prior to surgery; therefore, HbA1c was not measured. However, her preoperative fasting blood glucose level showed no elevation, suggesting that she was not diabetic at the time of surgery. We have added this information to the revised manuscript to clarify her preoperative metabolic status (lines 42-44).

Comments 2: You describe the surgeon placing their hand on the arm of the patient, leading to ‘excessive abduction of the arm’ (page 2 line 61-62). Could you please clarify which arm? I suspect it would be the left, give discussion of possible brachial plexus injury, but it is not explicitly stated.

Response 2: As the reviewer correctly inferred, the excessive abduction occurred on the left arm. We have updated the manuscript to explicitly state this detail Line 64).

Comments 3: Was the left radial arterial line placed under ultrasound guidance, or by palpation alone? You confirm that there was no clear vascular injury by ultrasound (page 2 line 68), but there is no mention of nerve (superficial radial/median) integrity. Was this checked?

Response 3: Thank you for raising this point. The left radial arterial line was placed using palpation and was successfully inserted on the first attempt. Additionally, during the postoperative ultrasound assessment, no signs of nerve swelling or other abnormalities suggestive of superficial radial or median nerve injury were observed. We have added this clarification to the revised manuscript (lines 59 and 72-74).

Comments 4: You mention a negative Tinel’s and Phalen exonerates carpal tunnel syndrome (page 2, line 72), but the sensitivity of these two maneuvers is quite low at 34-50% and 67-73%, respectively (see doi:10.1001/jama.2022.8481 AND doi:10.1093/ptj/pzad029). Was a nerve conduction/EMG completed?

Response 4: Thank you very much for this important comment and for pointing out the limited sensitivity of Tinel’s and Phalen’s tests for diagnosing carpal tunnel syndrome. We fully agree that these maneuvers alone cannot reliably exclude median nerve involvement.

In this case, a nerve conduction study or EMG was not performed. However, we acknowledge that electrophysiological testing would have been valuable for more accurate localization and characterization of the neuropathy. We have added this point to the Discussion section, noting that EMG/NCS should be considered in future similar cases to improve diagnostic precision ((lines 186-189).

Comments 5: In Figure 1, please indicate (by dotted line and text) which level is shown in axial view.

Response 5: We agree that this improves clarity. Figure 1 has been updated with a dotted line and text indicating the axial level shown.

Comments 6: In your discussion (page 3, line 99) you mention that symptoms were not due to the epidural, but then say that weakness improved after discontinuation. While I agree that there was evidence of a hematoma from the epidural anesthesia causing the symptoms, I would suggest clarifying this in the text. Clearly, if her weakness improved by stopping the anesthesia, then the anesthesia was causing the weakness (as expected), just not due to a hematoma.

Response 6:  Thank you for this important clarification. We agree that the wording in the original discussion could be misleading. The lower limb weakness improved after discontinuation of the epidural infusion, indicating that the weakness was caused by the pharmacologic effect of the epidural local anesthetic, rather than by an epidural hematoma or a procedure-related complication. We have revised the text to clarify that the symptoms were due to the expected motor effects of the local anesthetic and not due to structural injury from the epidural procedure (lines 109-113).

Round 2

Reviewer 2 Report

Comments and Suggestions for Authors

Thank you for considering my comments.

The authors have addressed my concerns.

Reviewer 3 Report

Comments and Suggestions for Authors

The authors patiently addressed my concerns, and I have not additional comments. There discussion of post-operative inflammatory neuropathies is a nice addition.